# Natural Cryoprotective and Cytoprotective Agents in Cryopreservation: A Focus on Melatonin

**DOI:** 10.3390/molecules27103254

**Published:** 2022-05-19

**Authors:** Giada Marcantonini, Desirée Bartolini, Linda Zatini, Stefania Costa, Massimiliano Passerini, Mario Rende, Giovanni Luca, Giuseppe Basta, Giuseppe Murdolo, Riccardo Calafiore, Francesco Galli

**Affiliations:** 1Department of Pharmaceutical Sciences, Lipidomics and Micronutrient Vitamins Laboratory and Human Anatomy Laboratory, University of Perugia, 06126 Perugia, Italy; giadamarcantonini94@gmail.com (G.M.); desiree.bartolini@unipg.it (D.B.); lindazatini94@gmail.com (L.Z.); 2Angelantoni Life Science S.r.l., 06056 Massa Martana, Italy; stefania.costa@angelantoni.it (S.C.); massimiliano.passerini@angelantoni.it (M.P.); 3Department of Medicine and Surgery, Section of Human, Clinic and Forensic Anatomy, University of Perugia, 06132 Perugia, Italy; mario.rende@unipg.it; 4Department of Medicine and Surgery, University of Perugia, 06132 Perugia, Italy; giovanni.luca@unipg.it (G.L.); gius.basta@gmail.com (G.B.); gmurdolo@tiscali.it (G.M.); riccardo.calafiore@unipg.it (R.C.); 5Centro Biotecnologico Internazionale di Ricerca Traslazionale ad Indirizzo Endocrino, Metabolico ed Embrio-Riproduttivo (CIRTEMER), 06132 Perugia, Italy

**Keywords:** cryopreservation, melatonin, cytoprotection, antioxidant, DMSO, stem cell, gametes

## Abstract

Cryoprotective and cytoprotective agents (Cytoprotective Agents) are fundamental components of the cryopreservation process. This review presents the essentials of the cryopreservation process by examining its drawbacks and the role of cytoprotective agents in protecting cell physiology. Natural cryoprotective and cytoprotective agents, such as antifreeze proteins, sugars and natural deep eutectic systems, have been compared with synthetic ones, addressing their mechanisms of action and efficacy of protection. The final part of this article focuses melatonin, a hormonal substance with antioxidant properties, and its emerging role as a cytoprotective agent for somatic cells and gametes, including ovarian tissue, spermatozoa and spermatogonial stem cells.

## 1. Introduction

Cryopreservation techniques are routinely used in a range of medical and research applications to store biological materials, including cells, tissues and their fractions. Unfortunately, even the most efficient protocols and technologies for cryopreservation are not able to guarantee complete control of the molecular processes responsible for the spontaneous deterioration of the biological specimens, starting from lipid autoxidation, which is the first to develop by the activity of molecular oxygen [1].

In cell and tissue samples, the cryopreservation process itself can interfere with the integrity of the stored material inducing cellular damage. The most important effects involve the cellular membrane and are the consequence of chemical and mechanical (mainly osmotic and ice crystal-induced) stresses occurring during the freezing and thawing steps of cryopreservation [2,3]. These include changes in lipid–lipid and lipid–protein interactions, membrane receptor function, signal transduction, permeability and transport mechanisms. Furthermore, as observed during the cryopreservation of biological fluids [1], lipid peroxidation represents a major event in cellular membrane damage during cryobanking of mammalian and plant cells [4,5], and is a trigger for other cellular alterations, including oxidative stress, DNA and protein damage, abnormal lipid signaling and activation of cell death programs (these aspects have been extensively reviewed elsewhere in [2,6,7,8,9,10,11] and references therein). 

Cryoprotective and cytoprotective agents (CPAs) are used to prevent these drawbacks of cryopreservation in either cellular or supracellular specimens. The ideal CPA should be able to prevent unwanted oxidation, the damaging effects of the cryopreservation process on membrane physical and chemical properties, and other cellular injuries introduced earlier and summarized in Figure 1. Cryoprotective agents (Table 1) act by reducing ice formation at any temperature by increasing the solutes concentration present in the system [12]. These agents are used to protect cell membrane integrity, avoiding damage to membrane lipids and consequently other cellular components, including cellular proteins and nucleic acids. These cellular effects exemplify the cytoprotection function of cryoprotective agents. However, other agents can be used during cryopreservation to protect cells and tissue samples that are not cryoprotective agents, thus acting to prevent cellular damage independent from their activity on ice crystal formation and osmotic effects. These include antioxidants that may prevent the oxidative stress effects of cryopreservation and critical constituents of specialized cells and tissues, such as ascorbic acid and chondroitin sulfate in cartilage tissue, respectively [13]. Melatonin (MLT) is another example of this type of CPA that has emerged in the last decades as a potent antioxidant and gene-modulation molecule, with a physiological role in the cytoprotection of a range of human cells and tissues [14,15].

The toxicity of most common CPAs limits their application in medical protocols of cryopreservation, including those intended for cell therapy and tissue/organ transplantation of immune hematology, regenerative medicine, and the reproductive biology area of competence. These aspects have extensively been described in other recent review articles (see for example [3,25,26,27]) and will find limited space in this mini-review article; for this reason, we apologize to all the authors that have not been cited in our article even if they have given important scientific contributions to this field. 

The most common CPAs used in cryobanking protocols are compounds of synthetic origin showing different levels of biocompatibility with impact on both cell physiology and the environment. At the cellular level, their toxicity depends on different mechanisms, including membrane effects, alterations of cell permeability and interactions with cellular proteins and mitochondrial function (Figure 2). 

Natural CPAs have become a popular subject among researchers that have attempted to find an alternative to synthetic CPAs [28,29,30]. However, much less is known about their mechanism of action and efficacy in cryopreservation. These include antifreeze and thermal hysteresis proteins, ice nucleation agents (INAs), sugars and natural deep eutectic systems (NADES), which are all described in this narrative review article as the main classes of natural CPAs, describing their alleged targets, protection mechanisms and possible side effects. Moreover, a particular focus is addressed regarding cytoprotection in cryopreservation, which is practically a neglected topic in the literature, and in this respect, we describe MLT and its emerging role as a para-physiological cytoprotective agent of somatic cells and gametes, including spermatozoa, ovarian tissue and spermatogonial stem cells (SSCs), that may be used for applications in cell therapy of male infertility. 

## 2. The Cryopreservation Process and the Role of CPA

Cryopreservation consists of the utilization of low temperatures to inhibit metabolism and retard biochemical processes responsible for the degradation of living cells and tissues. There are several approaches to this type of biopreservation, the optimum choice of which is dictated by the nature and complexity of biological material and the field of application (e.g., medical vs. scientific research) [31]. Better results are obtained with isolated cells in suspension, for which cell-specific cryobanking protocols and recommendations are available [28], whereas many more problems remain in the banking of tissues and organs. 

The main issues related to this process derive from the freezing of intracellular water (i.e., its transition from a liquid to a solid crystalline phase), which results in the formation of ice crystals that destroy cells through mechanical actions and changes in the composition of the liquid phase [2,32] (Figure 2). Vitrification represents an alternative to freezing-induced cellular damage, leading to solidification of cellular water to a glass-like state without ice formation, (recently reviewed in [33]), but its application is usually limited to small volumes, while freezing can be used for a wide range of sample volumes. Osmotic and oxidative stress are other important aspects of cellular damage during cryopreservation and freezing–thawing cycles. To circumvent these problems, the addition of CPAs is a necessary step.

Cryoprotective agents are classified into two main categories, namely: nonpenetrating (NPA) and penetrating agents (PA), that protect biological samples through different mechanisms (Table 1). 

Nonpenetrating agents are small molecules (e.g., sugars) organized into polymers that increase extracellular osmolality to cause cell dehydration as a stabilization mechanism; because of this function, nonpenetrating agents require slow cooling rates. These agents are added to thawing media to slow down the influx of water in the cell, thus preventing osmotic shock and lysis [34,35]. In this context, ice recrystallization is an important process to control during cryopreservation in order to prevent cold-induced damages. This is a form of Ostwald ripening, where larger ice crystals are formed at the expense of smaller ones, which causes osmotic stress and even cell death (reviewed elsewhere in [36]). NPA can be used to prevent the damage of ice recrystallization, including for example polyvinyl alcohols and polyampholyte-based CPAs, which are synthetic polymers of different sizes and levels of biocompatibility. These have been demonstrated to protect cells with different mechanisms, such as the induction of increased viscosity, glass transition facilitation, and water and salt trapping in the polymeric matrices, preventing intracellular ice formation and osmotic shock during freezing [18]. Polyampholytes have also been reported to cryopreserve cells by strongly interacting with the cell membrane, with hydrophobicity increasing the cytoprotective effect by increasing both the affinity for membrane interaction and affinity for ice recrystallization inhibition [19]. This type of NPA can be of either synthetic [19,20] or semisynthetic nature, such as carboxylated ε-poly-ʟ-lysine polymers [18] that have successfully been implemented into the cryopreserved human mesenchymal stem cells and induced pluripotent stem cells. In addition, temperature-responsive physical gels and zwitterionic polymers have been used in the cryoprotection of 3D cell cultures in which these materials can act at the same time as the 3D scaffolds and CPA in the composite cell system [21].

Penetrating CPAs are small and nonionic molecules (e.g., sulfoxides, alcohols, amides and imides) that can cross the cell membrane to exert multiple and partially unknown mechanisms of action (these are further discussed in Section 3); these CPAs can interact for instance with cellular components such as enzymes and signal transduction proteins, but at the same time, they are known to be more toxic compared to the nonpenetrating ones [12]. 

CPA systematics may also include the categorization of synthetic and natural agents (presented in Section 4), macromolecular and small-molecule CPAs, antifreeze solutes and ice blockers, vitrification agents, toxicity neutralization agents, and methoxylated and hypertonic CPAs [3,33].

## 3. Challenges in the Utilization of CPAs

As introduced earlier (Section 1 and Section 2), CPAs are used to mitigate the negative effects of cryoinjury [28]. Their mechanisms of action vary depending on the type of CPA and cryopreservation protocol. The most common CPAs used in standardized cryopreservation protocols include glycerol and dimethyl sulfoxide (DMSO). Usually, in cell cryopreservation, these CPAs are dissolved in hypertonic media in the molar concentration range [27,37]. These are described as multi-modal agents since they promote multiple effects and actions on specific and nonspecific targets. Primarily, they act through osmotic dehydration of the cells that prevents ice crystal formation. They also play an important role in the thawing process, avoiding ice crystals that may recrystallize into bigger ones, thus leading to cell death [38], a process discussed earlier for NPA (see Section 2). In addition, glycerol and DMSO have long been described to interact with polar head groups of phospholipids to increase membrane stability [39]. Moreover, investigation of freezing injury to the cell membrane and cell protein components have demonstrated that DMSO is able to decrease membrane conformational disorders during freezing at −40 °C and delay protein denaturation in the warming process [40]. Glycerol facilitates the osmotic flux of water out from the cells during progressive freezing of the cell samples [41]. Collectively, these studies and many others in the literature demonstrate the ability of these CPAs to interact at sub-zero temperatures with membrane phospholipids to mitigate membrane lipid-phase transition and to protect protein structure.

However, these CPAs show major adverse effects, possibly by the multiple interactions with intracellular components and functions, including enzymatic reactions, membrane transporter mechanisms and ion exchange [27,42]. During slow cooling protocols, these CPAs are applied at molar concentrations, resulting in significant amounts of the agent that accumulates within the cell [28]. Then, diluting post-thawed cells directly into the isotonic culture medium results in a significant water uptake by the cells causing osmotic stress and disruption of homeostatic mechanisms. 

The ability of most CPAs, such as DMSO, to reach different intracellular compartments can damage the cell through several mechanisms, including defects of glycolytic energy production, oxygen consumption, and electrolyte balance [42]. Other mechanisms of cellular damage by these CPAs have been alleged to include effects on cellular protein stability and function. Arakawa et al. [43] studied CPA toxicity by disruptive protein interactions. Their studies showed that at supra-zero temperatures, DMSO and ethylene glycol destabilize proteins, forming hydrophobic interactions, while at sub-zero temperatures, they stabilize them.

Organelle-specific toxicity has been hypothesized and confirmed in several studies, demonstrating how oxidative damage is an important consequence of these CPAs (recently reviewed in [34]). This is the case for DMSO, for example, which remains the CPA of choice in the cryopreservation of mammalian cell lines [27,28]. This membrane-penetrating agent can be dangerous for the cells, as it can cause alterations of the mitochondrial membranes, thus increasing electron leakage and the production of reactive oxygen species (ROS). To avoid these adverse effects, the agent has to be removed as soon as possible, particularly after the cells are thawed [44]. Dimethylsufoxide toxicity is also a concern in cell therapy protocols. In patients receiving infusions of cell-therapy products treated with this CPA, significant side effects have been reported, including cardiovascular, neurological, gastrointestinal, and allergic reactions, and DMSO has been associated with hematological disturbances as an altered expression of natural killer (NK) and T-cell markers in circulating leukocytes and their in vivo function [27]. These aspects are a major limit to the use of DMSO-based CPAs in the manufacturing of autologous and allogeneic cellular therapy products such as chimeric antigen receptor (CAR)-T and CAR-NK cells.

Recently, several studies were published on the epigenetic modifications induced by CPA. It is well known that the exposure to even low concentrations (1% or less) of DMSO, can cause alterations in the epigenetic profile of embryonic stem cells. Genomic effects have been identified in the cryopreservation of gametes as well [45], although these may not only depend on CPA exposure [46]. Therefore, this area of CPA toxicity is worth investigating, as more accessible methods to study epigenetic mechanisms will become available.

Strategies to minimize CPA toxicity have been proposed, including the combination of chemical agents with different levels of toxicity (such as acetamide and formamide) with less toxic or neutralizing CPAs, and the use of more biocompatible and natural CPAs, some of which have been identified to act as toxicity-minimizing agents. These aspects are presented in the next section (Section 4). 

## 4. Natural Cryoprotective and Cytoprotective Agents

The possibility to use natural CPAs in cryopreservation has stimulated interest in the scientific community, representing an alternative to synthetic counterparts and their limits in terms of toxicity and cryoprotection function. As introduced earlier in Section 1, these include cryoprotective agents presented in this section, such as antifreeze proteins (AFPs), INAs, sugars and NADESs, which will be presented in more detail in this section.

Natural agents with cytoprotection function have also been utilized in cryopreservation, including antioxidants and cell stress preservation agents, such as ascorbic acid [13], vitamin E (as α-tocopherol) [7], glutathione [4] and phenolic substances such as quercetin [5], MLT (*vide infra*, Section 5) and others (reviewed in [11]). This specific group of agents respond to the need of neutralizing the pro-oxidant effects of cryopreservation (Figure 1), but it may also be important to mitigate the toxicity of other CPAs (Figure 2). For instance, in the case of vitrification protocols that use high concentrations of toxic CPAs, the antioxidant and detoxification metabolite glutathione [47] was added to the freezing medium to control lipid peroxidation [4]. Melatonin has also been used to mitigate the toxicity of other CPAs in gamete cryobanking (discussed in Section 5). The same “toxic CPAs” can be used with this aim, combining certain CPAs that can directly neutralize the toxicity of other CPAs (reviewed in [3,48]); this is the case, for example, for DMSO, which can neutralize the toxicity of formamide, allowing for a higher total concentration of these CPAs to be used with reduced toxicity. Ethylene glycol can be used in vitrification solutions in place of polyethylene glycol to reduce the nonspecific toxicity of the solution, allowing for increased hydration and protection of cellular macromolecules [34,49]. In addition, sugars have been demonstrated to behave as toxicity-minimizing agents [34] (further described in Section 4.2). 

### 4.1. Antifreeze Proteins (AFPs)

During evolution, living organisms have developed strategies to survive in cold environments, including antifreeze proteins (AFPs) and ice nucleation agents (INAs) [29]. 

Antifreeze proteins and glycopeptides AF(G)Ps are part of a family of natural CPA used by animals, such as insects, amphibians and fish, to limit ice formation and to protect against “freezing hysteresis” (i.e., difference between the equilibrium melting point and the nonequilibrium freezing point). Whether AFPs exert a protective role as cryoprotectants depends both on their concentration and the type of selected AFP. 

In Antarctic fish, the presence of AFPs protects them from freezing below −2 °C; this is also the case in insects and in freeze-tolerant plants found at sub-zero temperatures [50]. The nonequilibrium freezing point is the temperature for which a crystal in a supercooled solution suddenly grows [24]. In the cryopreservation of cell samples, this is part of a process identified as ice recrystallization (further described in Section 2), which represents a major cause of cell dehydration, membrane rupture, and subsequent cryodamage. As a consequence, the family of peptides, glycopeptides and proteins included in the definition of AFPs has been identified as promising CPAs that are able to act by non-colligatively depressing thermal hysteresis, which is responsible for ice crystal equilibrium and for the inhibition of cell damage. Research concerning the mechanism of action for AFPs suggests a direct binding of the proteins at the ice prism surface, which prevents ice crystal growth through a specific water molecule arrangement with the surrounding ice [51]. 

These properties of natural AFPs have recently inspired the development of new ice-interactive cryoprotectants that have been used as an alternative to DMSO in the cryopreservation of human hematopoietic stem cells [52].

In extreme cold environments, nature has provided additional tools to promote ice formation through ice nucleating agents. In some insect species, the presence of INAs has been associated with a seasonal adaptive elevation of their super cooling points, which has inspired use of these compounds as a CPA. These agents favor ice formation even at extreme sub-zero temperatures [53]. Similar events are typical for some cold-tolerant amphibians species and marine mollusks [54]. 

### 4.2. Sugars

Several organisms (such as frogs, bacteria, yeasts and plants) live in different natural environments and efficiently use sugars (such as glucose, trehalose and raffinose) to survive in extreme climatic conditions, e.g., extreme droughts or freezing. 

Mammalian cell membranes are impermeable to sugars; thus, these molecules are used as extracellular additives. Different approaches have been attempted to overcome membrane permeability, such as thermotropic lipid-phase transition, transfection, poration [55], and even cellular microinjection [56]. 

Nonreducing disaccharides (trehalose) and trisaccharides (raffinose) can afford cytoprotection, and they constitute a valid alternative to standard CPA agents. According to this, Peterenko and colleagues [57] demonstrated that it is possible to partially substitute DMSO with sucrose in the cryopreservation of a range of cells, such as human fetal liver hematopoietic stem cells and human mesenchymal stromal cells. Moreover, monosaccharides (such as galactose) and disaccharides (such as sucrose or trehalose) have also been reported to act as toxicity-minimizing agents [34] that are worth investigating to develop more biocompatible combinations with other toxic CPAs. 

Sugars exert their actions by inducing lipidic membrane stabilization as a result of direct interactions with polar components of the bilayer and by preventing intracellular protein aggregation. In addition, they act as osmolytes to prevent osmotic stresses [58]. 

In contrast, smaller sugars such as glucose or its polymers, such as dextran, cannot provide optimal cryopreservation due to the inability to directly interact via hydrogen bond formation with the polar heads of cell membrane phospholipids [59]. 

Eroglu et al. [56] investigated the ability of raffinose to afford cytoprotection in mammalian oocytes when used in combination with DMSO. Raffinose was introduced inside the cells through a microinjection technique. The study proved that this nonreducing trisaccharide exerts an important cytoprotective action if combined with DMSO. The presence of raffinose in the intra and extracellular compartments reduced the concentrations of the penetrating CPA, providing optimal cryopreservation conditions. In the same study, the combination of intra- and extracellular raffinose with 0.5–1.0 M DMSO used for the slow cooling protocol had a beneficial effect on cryosurvival of oocyte, and embryonic developmental rates were maintained at levels comparable with those of the unfrozen samples [56]. This confirms the efficacy of sugars when used in combination with small amounts of other CPA, providing an important solution to minimize their toxicity.

Cryopreservation techniques, as a final purpose, have to safely bring cells below the glass transition temperature, at which the surrounding medium vitrifies. A high glass transition temperature can facilitate the cryopreservation procedure. Sugars are effective glass formers compared to conventional cryoprotection agents, such as DMSO, ethylene glycol and 1,2-propandiol.

### 4.3. Natural Deep Eutectic Systems (NADESs)

Deep eutectic systems (DESs) are characterized by high biodegradability, cost-effectiveness and low toxicity. DESs can originate from natural primary metabolites such as sugars, amino acids, organic acids or choline derivatives, which are mixed at a specific molar ratio. These systems have been “bioinspired” by the observation that these metabolites are produced by animals and plants to adapt in extremely cold conditions. The mixtures of these components have a higher melting point depression compared to the individual constituents, such that they become liquid at room temperature [60]. 

Trehalose, glucose, sorbitol and proline constitute the components of DESs, even if other compounds have recently been used to develop new solvents such as naturally occurring l-ascorbic acid (vitamin C) and choline [61]. 

These molecules are successfully used in combination with DMSO to improve cell viability after thawing. The mixture of trehalose with DMSO has demonstrated to be effective in the cryopreservation of several cell lines, such as stem cells, primary hepatocytes, and HepG2 cells [60].

NADES have been recently explored as potential CPAs for the preservation of different types of cells, including lactobacillus [62], mouse fibroblast cells [63], mesenchymal stem cells [64], and Jurkat cells, [30], as well as in the development of DMSO-free protocols of cryopreservation of NK and T cells, that remains a challenge for the development of autologous and allogeneic cell therapy products of these lineages, such as CAR-T and CAR-NK cells (recently reviewed in [27]). In this respect, in Jurkat cells, an immortalized cell line used as a T lymphocyte model, a multicomponent osmolyte solution composed of the naturally occurring metabolites trehalose, glycerol, and isoleucine provided 84% post-thaw recovery that was comparable to the results obtained using 10% DMSO [65]. In these studies, the activity of individual osmolytes in reducing the damaging intracellular ice formation was much lower compared with that of an optimized combination of these solutes. Moreover, interactions between specific solutes were observed that may influence post-thaw recovery in a nonlinear manner. More in detail, interactions between sucrose and glycerol and sucrose and isoleucine were found to improve post-thaw recovery, and a similar effect was obtained in increasing glycerol concentrations.

Another important consideration is that the cryoprotective effect of NADES composed of different sugars combinations (such as trehalose–glucose, glucose–proline, glycerol–glucose, etc.) can vary in a cell-dependent manner. For example, after thawing–freezing cycles in L929 cells, NADES produced almost the same cytoprotection effect of the standard CPA DMSO, while in HaCaT, these promoted a recovery of cell viability that was significantly higher compared to DMSO [60]. 

NADES represent a great alternative to CPAs in the cryobiology field, because during the thawing process, they do not need to be removed from the freezing media.

## 5. The Role of Melatonin (MLT) in Cytoprotection and Cryopreservation

In addition to its canonical function as a circadian rhythm regulator, MLT has been fully demonstrated to represent a potent CPA and cell homeostasis factor [15,66]. This hormone is primarily synthesized in the pineal gland starting from the essential dietary amino acid tryptophan, but it is also produced in other tissues, such as the ovaries and testes. The properties of this molecule explain its beneficial effects on different biological functions and clinical aspects [67]. The cytoprotective mechanisms of MLT are linked to its pleiotropic nature of antioxidant, anti-apoptotic, anti-inflammatory, and autophagy modulation agents [68]. 

Melatonin is considered to be a potent repressor of cellular oxidative stress via its activity as a direct antioxidant and free radical scavenger that depends on its hydrogen atom transfer properties [69], and via its capability to bind transition metals [70]. In addition to this, MLT regulates gene expression; the activity of antioxidant enzymes, such as glutathione peroxidase, superoxide dismutase and catalase [69]; stress and survival MAPKs; and stress response transcription factors [15,71]. These functions may explain the efficacy of MLT in preventing cryopreservation-induced oxidative stress and the loss of viability and physiological functions in the stored cells [69] (Figure 3).

The antioxidant actions of MLT are linked to its interaction with receptors and transcriptional factors located on either the cell membrane or subcellular organelles [15,72,73]. This process triggers the induction of antioxidant and cytoprotection genes, such as GPx, SOD 1,2, and SIRT3. However, the mechanism by which MLT or its metabolites modulate the expression and activity of cytoprotection genes has yet to be fully understood. One proposed mechanism is its ability to activate Nrf2 transcriptional function by the inhibition of Nrf2/Keap1 complex ubiquitination and proteasomal degradation. These effects promote the translocation of Nrf2 into the nucleus, where it binds to antioxidant and electrophile response element (ARE/ERE) sequences on the promoter region of the target genes to induce their transcription [14].

In the mitochondria, the enzymatic dismutation of superoxide anions is facilitated by the stimulation of SIRT3, leading to the deacetylation and activation of SOD2. This process limits the damage to the organelle [72]. 

At the same time, the interaction with the reactive species metabolizes MLT into cyclic-3-hydroxymelatonin, N1-acetyl-N2-formyl-5-methoxykynuramine (AFMK) and N1-acetyl-5-methoxykynuramine (AMK), having similar or better detoxifying actions compared to their precursor [73,74].

Therefore, whereas classical antioxidants scavenge only a single ROS, MLT can detoxify multiple species via its direct scavenging activity and transcriptional effects. 

These properties of MLT can be important to afford cytoprotection in the cryopreservation of somatic cells. Once added to freezing solutions, MLT can easily cross the cell membrane to exert its biological function, including antioxidant and stress response effects, mitochondrial protection, and control of superoxide anion formation [75]. These effects have preliminarily been investigated by Solanas et al. in primary human hepatocytes [76], for which cryopreservation represents a fundamental tool to guarantee sufficient availability for both clinical and research applications, including an extracorporeal bioartificial liver, a technique used to treat liver disease patients and to vicariate orthotopic liver transplantation [77]. In these studies, MLT was confirmed to be a CPA, but its efficacy was lower compared to DMSO [78]. 

The most striking evidence regarding the cytoprotective function of melatonin has been reported in the case of gametes cryopreservation, which is presented in the next sections.

### 5.1. Ovarian Tissue

The aim of ovarian tissue cryopreservation is the maintenance of female fertility, especially in the case of patients who undergo radiotherapy or chemotherapy that damage the ovaries, causing premature loss of ovarian functions. The preservation of ovarian tissue (OT) is the most feasible option to attempt preservation of fertility in adolescent cancer patients for whom the cryopreservation of oocytes and embryos is impossible [79].

Because the cell types present in this specific tissue are vulnerable to severe damage, cryopreservation of OT faces different challenges. Mainly, the production of ROS causes oxidative damage and the disruption of follicular functions. ROS production has been reported to affect the vitrification process, causing cell dysfunction and a decline in oocyte survival rate by the activation of apoptotic cell death [80]. MLT has been used in the cryopreservation of OT from adolescent SPF KM mice to counteract the adverse effects of oxidation on follicles [81]. Treated animals presented a well-preserved follicular morphology, with increased dimensions compared to the untreated ones, and a dose–response effect of MLT was observed when there was a decrease in apoptotic follicles, suggesting that this hormone may also play a role in the development of polycystic ovaries. In this study, MLT was found to interact with Nrf-2 activity and hemeoxygenase-1 (HO-1) expression, and the authors proved that the use of MLT during the vitrification process enhances resistance against cryopreservation damage, inducing the expression of the antioxidant enzymes superoxide dismutase, catalase and GSH-Px as well as of GSH and T-AOC. Consistent with the anti-apoptotic activity observed in follicles, MLT also induced the expression of the mitochondrial protein Bcl-2. 

Studies on the cryopreservation of OT deriving from adolescent Sprague–Dawley rats confirmed this effect of MLT in protecting the integrity of follicular morphology [82], with a number of intact follicles and apoptotic follicular cells that were reduced during the freezing–thawing process. The levels of apoptotic proteins, especially Bax levels (pro-apoptotic protein), decreased due to the effect of MLT in the cryopreserved OT, with a corresponding increase in Bcl-2 levels (anti-apoptotic protein). Furthermore, MLT promotes the expression of Nrf2 mRNA in cryopreserved oocytes, suggesting an underlying role of this transcription factor in the cytoprotection function of the molecule. Nrf2 downstream genes involved in this response to MLT include HO-1, CAT, SOD and GSTM1, which are important players in stress adaptation mechanisms of the cell [83].

### 5.2. Sperm Cells

Cryopreservation is known to induce oxidation processes in spermatozoa, which are the main mechanisms causing cryodamage and reduced motility and viability in these cells [84]. These defects are responsible for male infertility, and thus, their development must be prevented during cryopreservation. Sperm motility is dependent upon mitochondrial function, and mitochondria represent a critical site for the generation of oxidative stress [7,85]. Abnormal ROS concentrations can disrupt the coupling between electron transport and oxidative phosphorylation, resulting in the loss of mitochondrial function and consequently in reduced sperm motility. Oxidative stress may also cause apoptotic cell death in cryothawed spermatozoa, as demonstrated by the investigation of phosphoserine externalization to the outer leaflet of the plasma membrane and mitochondrial depolarization. Furthermore, ROS overproduction can have damaging effects on the DNA in the sperm nucleus [84], with levels of DNA fragmentation that linearly correlate with the levels of ROS detected in the gametes [85]. 

In a recent meta-analysis, MLT was included among the antioxidants that are able to reduce ROS production and protect human sperm function during the freeze–thaw process [86]. This molecule may protect sperm parameters during cryopreservation through different mechanisms. Najafi et al. [87] investigated the effects of MLT on human sperm oxidative damage and fertilization parameters, including motility, viability and levels of intracellular ROS (H_2_O_2_ and O^2−^). In this study, MLT used at a concentration of 3 mM significantly reduced the levels of caspase-3 and the percentage of apoptotic and nonviable sperm cells, also increasing the motility and levels of AKT phosphorylation of sperm cells. Again, MLT can protect mitochondria through different mechanisms [88]. These include direct antioxidant and free radical scavenging effects and an enhanced activity of the respiratory complexes C-I, C-II and C-IV, which are important in ATP production. Melatonin has been described to reduce electron leakage from these complexes, which is responsible for mitochondrial generation of superoxide anions. In addition, it is reported to reduce apoptotic signaling and to protect the cell membrane from free radical damage in spermatozoa, thus preserving membrane fluidity and the percentage of viable gametes [89]. 

The role of MLT as a useful CPA in sperm cryopreservation has been confirmed in a series of recent studies in different species, including fish [90] and dog [91]. 

Despite the evidence of an enhanced resistance to cryodamage in spermatozoa, the available information on dosages and protocols for the use of MLT remains insufficient for a translation of this CPA into clinical guidelines of reproductive medicine for sperm cryobanking. In addition, mechanistic aspects through which MLT can positively affect human and animal spermatozoa during cryopreservation are worth investigating further.

## 6. Melatonin in Stem Cells Cytoprotection

The protective effects of MLT have extensively been studied in stem cells exposed to different types of cellular stressors, including heavy metals and xenobiotics [15,66,92]. Little is known about MLT cytoprotective effects during stem cell cryopreservation. The available literature in this area mainly reports on spermatogonial stem cell (SSC) cryopreservation, which is a relevant aspect in reproductive biology presented in this section. 

### Spermatogonial Stem Cells (SSCs)

Spermatogonial stem cells (SSCs) are the base ground for spermatogenesis and male fertility. They are a type of primitive spermatogonia located in the seminiferous tubules [93]. SSCs are able to both self-renew, to keep the stability of the stem cell pool, and differentiate, contributing to spermatogenesis maintenance and the homeostasis of testicular cells [92]. These cells can be maintained in culture, and after transplantation, they can be differentiated into sperm cells [94]. The number of SSCs appear to be lower compared to other testicular cells, approx. 0.02–0.03% of mouse testicular cells. Therefore, the cryopreservation of these cells has gained great credit in the field of early reproductive biology and medicine [95,96].

When the protective effects of MLT were studied in cryopreserved goat SSCs [97], a 10^−6^ M concentration of this molecule was found to significantly increase cell viability, total antioxidant capacity (T-AOC) and the mitochondrial potential of these cells, while the levels of ROS and malondialdehyde (MDA) were reduced. Melatonin also improved enzyme activity and the expression of SOD, CAT and GSH-Px. At the mitochondrial level, the expression of the pro-apoptotic protein Bax was inhibited, and the expression of its anti-apoptotic counterpart Bcl-2 increased; moreover, MLT reduced both mitochondrial swelling and vacuolation, inhibiting the release of cytochrome C into the cytosol and inhibiting caspase-3 activation, which are important events in the mitochondrial pathway of SSC apoptosis. Again, MLT reversed the mechanisms of freezing-induced autophagy in SSC, regulating the expression of the autophagy-related proteins LC3-I, LC3-II, P62, Beclin1 and ATG7. 

According to these findings in goat SSCs, MLT was found to promote beneficial effects in terms of cell proliferation and survival rates in mice SSCs [98]. 

These studies and others in the literature suggest that the implementation of MLT in SSC cryopreservation may represent an important strategy in the field of assisted reproduction, considering either an application in future auto-transplantation, or the in vivo induction of mature sperm development [99].

## 7. Conclusions

Cryopreservation can be affected by mechanical and chemical complications that influence the quality and viability of the preserved cells or tissues. Therefore, the use of CPAs is fundamental to ensure a safe process. These agents exert their cryoprotective function by increasing the total concentration of all solutes in the system, thus reducing the amount of intracellular ice formation. To exert this function, the agents should be able to penetrate cells and ensure low cytotoxicity. Still, due to DMSO’s well-proven high efficacy in cryoprotection, this remains the most used CPA in biomedical research. However, as an organic solvent, DMSO has an intrinsic toxicity and can induce cell apoptosis and uncontrolled differentiation, even if used at low dosages.

New techniques of cryopreservation have been developed and implemented in recent years, including CPA combinations [100] and new CPA techniques of application. For instance, to increase post-thawing viability, cells can be encapsulated with alginate prior to freezing, a procedure that has successfully been applied in human hepatocytes through microbead encapsulation [28].

Furthermore, as highlighted in this review article, natural CPA can represent an alternative to DMSO and other classical CPAs of synthetic origin, reducing their toxicity during cryopreservation. Due to these properties, these agents have gained increasing attention in the last several years. These include AFPs that hold great potential for an application in cryobanking [29]. Although these proteins and peptides can have immunogenic effects and are difficult to produce, they represent an alternative to synthetic CPAs. Saccharides, such as fructose, galactose and glucose, are found in plants, fishes and insects exposed to ultralow temperatures. They can limit intracellular dehydration in the process of freezing, thus preserving the cellular membrane structure and its functions [30]. Trehalose, a natural nonreducing disaccharide, has shown good performance in cryopreservation, but due to its poor membrane permeability, it should be delivered into the cells through different strategies such as microinjection, pore formation or internal trehalose synthesis using genetic engineering techniques. Again, NADES have been described to represent an alternative to CPAs in the cryobiology field. These CPAs do not need to be removed from freezing media during the thawing process, and promising positive results have been obtained using them in the cryoprotection of different types of cells, including NK and T cells for which DMSO-free cryopreservation protocols will be important to scale up autologous and allogeneic cellular therapies based on these cell lineages, such as CAR-T and CAR-NK cell therapy [27]. 

Neutralizing the toxicity of CPAs is an important aspect in cryopreservation. This can be achieved with classical CPAs that have specific neutralizing activity against other and more toxic CPAs that are used in combination, e.g., DMSO was found to partially neutralize formaldehyde toxicity [3]. Alternatively, specific cytoprotective agents can be used. These include MLT, which has been demonstrated to be efficient in neutralizing free radicals and other reactive intermediates responsible for oxidative stress and cytotoxicity during cryopreservation. In this review article, we described these cytoprotective effects of MLT and their application in the field of reproductive biology, with a series of promising results obtained in the cryoprotection of ovarian tissue, spermatozoa and SSCs. A better understanding of the chemical and biological mechanisms underlining the freezing process will be crucial for a more efficient use of this and other toxicity-mitigation agents, also expanding their applications in scientific research and medical protocols.

In perspective, genetic studies have suggested the possibility of monitoring the susceptibility of the cryopreserved material to cryoprotectant toxicity [101]. This indicates an opportunity to optimize cryopreservation protocols, driving them in the era of precision medicine. Practical implications may include a better selection of CPA and the development of genetic and pharmacological tools to improve CPA function and cryopreservation protocols. Moreover, besides choosing the right CPA and developing new and more efficient ones, an important aspect for successful cryopreservation is to monitor CPA effects throughout all critical steps of sample storage. In agreement with these objectives, a recent generation of smart freezers has been developed to abate risks related to liquid nitrogen handling and operator intervention, as well as sample mishandling and contamination. These fully automated systems are designed to allow for complete control of the cryopreservation process, and they may pave the way to further levels of optimization in CPA selection and implementation protocols.

## Figures and Tables

**Figure 1 molecules-27-03254-f001:**
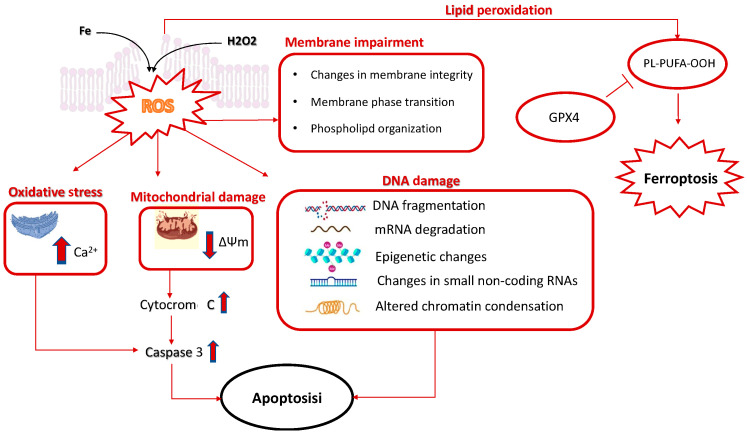
Membrane alterations caused by cryopreservation. The cryopreservation process can cause alterations of the cellular membrane and specific changes in phospholipid organization and bilayer integrity, as well as modifications of the membrane ultrastructure. The oxidative damage of membrane lipids caused by reactive oxygen species (ROS) overproduction is a key underlying event in cellular damage consequent to cryopreservation. An excess of free radicals can derive, for example, from the Fenton reaction or other cellular oxidative stress processes that ultimately sustain lipid peroxidation and membrane impairment. These include an excess of Ca2^+^ influx into the cytoplasm from the extracellular environment and from the endoplasmic reticulum. Moreover, oxidative stress induces a rapid depolarization of the inner mitochondrial membrane potential and subsequent impairment of oxidative phosphorylation, as well as the release of cytochrome c, which is a main trigger of the intrinsic pathway of apoptotic cell death. Membrane lipid peroxidation may lead to the formation of eicosanoids associated with ferroptotic signaling. Ferroptosis is an iron-dependent cell death program that is prevented by the activity of glutathione peroxidase 4 (GPx4) [16].

**Figure 2 molecules-27-03254-f002:**
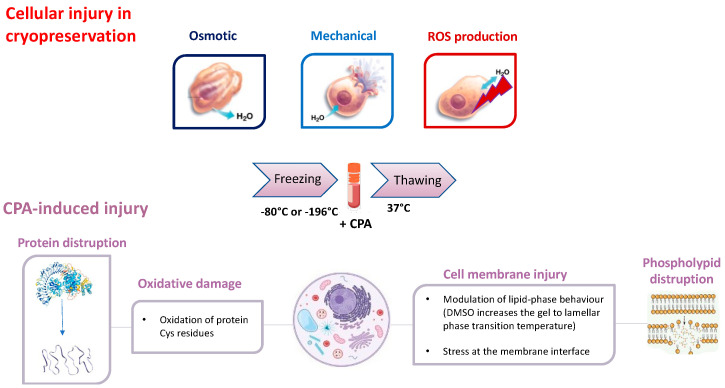
Cryopreservation and CPA−induced cell injury. When cryopreservation is not carried out under optimal conditions, the cell can undergo a series of alterations. These can be the result of osmotic, mechanical or ROS-induced injuries. The osmotic injury is caused by cell dehydration, while the mechanical injury is due to the formation of ice crystals within the cell that cause its disruption. Finally, ROS overproduction triggers the suppression of antioxidant mechanisms and the oxidative damage of cell components, such as membrane lipids and nucleic acids. The utilization of CPA prior to the freezing process, can prevent these injuries, but at the same time, it can interfere with membrane proteins and phospholipid stability and function.

**Figure 3 molecules-27-03254-f003:**
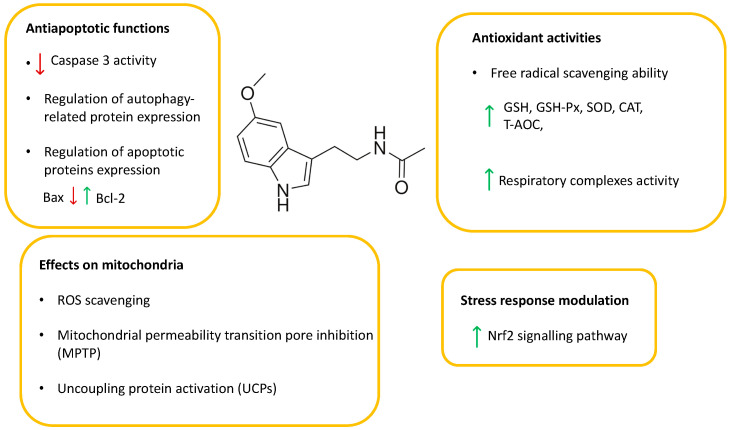
Melatonin cytoprotection function. Melatonin protects the cell from oxidative stress by inducing the expression of antioxidant genes (up arrow) such as glutathione peroxidase (GSH-Px), superoxide dismutase (SOD), catalase (CAT) and by increasing the total antioxidant capacity (T-AOC) in the presence of a stimulation of respiratory complexes activity. These effects are mediated by the stimulation of transcriptional proteins, such as nuclear factor erythroid 2 (Nrf2), the activity of which is particularly important for the CPA and stress adaptation effects of melatonin. Melatonin exerts antiapoptotic effects by decreasing (down arrow) Caspase 3 and Bax activity and by increasing (up arrow) the expression of B-cell lymphoma protein 2, e.g., Bcl-2. Other effects include the modulation of autophagy-related proteins. Moreover, melatonin acts directly on mitochondria, acting as an ROS scavenger, inhibiting the activity of MPTPs proteins, and activating UCP proteins. Upward and downward arrows correspond to “increased” and “decreased” expression or function of the reported parameters, respectively.

**Table 1 molecules-27-03254-t001:** Examples and characteristics of the most common cryoprotective agents.

CPA	Source and Examples	Classification	Reference
**Sulfoxides**	Synthesized from dimethyl sulfide (the prototypal CPA of this category is dimethyl sulfoxide)	Penetrating	[17]
**Diols**	Propylene glycol (or 1,2-propanediol) is the prototypical agent of this category of CPA that is produced via hydration of propylene oxide; other diols are methylene glycol and ethylene glycol	Penetrating	[17]
**Synthetic and semisynthetic polymeric agents**	Synthetic resins obtained from hydrolysis of polyvinyl acetate (such as polyvinyl alcohols), and copolymers and semisynthetic polymers of different origin, including polyampholytes (such as carboxylated poly-ʟ-lysine and others)	Nonpenetrating	[18,19,20,21]
**Saccharides**	Fungi, plants, invertebrate animals (examples are: sucrose and trehalose)	Nonpenetrating	[22,23]
**Proteins**	Animals (antifreeze proteins)	Nonpenetrating	[24]

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
