# Peer review of "Natural Cryoprotective and Cytoprotective Agents in Cryopreservation: A Focus on Melatonin"

_molecules, 2022, doi:10.3390/molecules27103254_

Round 1

Reviewer 1 Report

The manuscript by Marcantonini et al. discusses the various aspects of cryopreservation with a focus on melatonin. They have discussed several mechanistic aspects. Although this manuscript may be important, considering the topics covered, however, many aspects are not well-written. This manuscript may be considered for publication; however, an extensive revision should be done, and the manuscript should be checked very carefully before resubmission. Also, thorough proofreading of the manuscript should be done as there are many issues. Below are my comments.

  1. The literature survey is up to date. P3 L81, they mentioned that “Non penetrating agents per-81 form their function by increasing the osmolality of the extracellular environment”, although this is correct, however studies using polymers have been done and some more mechanistic details have been reported (e.g., Nature Communications volume 12, Article number: 1323 (2021), Biomacromolecules 2016, 17, 5, 1882–1893, Communications Materials volume 2, Article number: 15 (2021)). There are other reports also. These papers need to be discussed when summarizing non-penetrating agents.
  2. There are several problems with Table 1.

a. If you mention sulfoxides as a different class, then there should be more than one compound in this class.

b. What is ethanol homopolymer? PVA is a kind of ethanol homopolymer. PVA and its full form are mentioned separately.

c.  Ethylene glycol is a well-known permeating CPA. So, I believe 1,2-methylene glycol should also be permeating, though it is listed as non-permeating. Please add a reference to support this. And for other compounds also, references should be added to support the claims.

d. I do not think all the compounds listed in the Propylene glycol category can be classified as propylene glycols.

e. Why is Trehalose not listed with Saccharides?

f. Add a footnote in the Table and explain what the things are written in brackets. In some places, IUPAC names are given and in other places, common names are written. Please unify it.

g. References for everything should be added as a separate column in the Table for each category.

  1. I do not understand why two names for dimethyl sulfoxide are used. DMSO and Me2SO. It should be unified.
  2. The full form of ROS is not mentioned anywhere.
  3. It is better to define and differentiate cryoprotection and cryoprotection. And it is better to discuss if these are similar terms and if all CPAs are cytoprotective also? Can these two terms be used interchangeably universally? Because in this paper, the terms cryoprotective and cytoprotective are used randomly, without any explanation. For example, the last paragraph of P4 and the first paragraph of P5.
  4. P6 L235, Write the full form of NADES or write NADES in brackets in the subsection title.
  5. P7, L249, the use of the word “alleged” is inappropriate here.
  6. In the conclusion section, it would be better to include a future outlook and provide some discussion about what can be done in the near future to fill research gaps.

Reviewer 2 Report

In the review manuscript entitled “Natural cytoprotection agents in cryopreservation: a focus on melatonin” Authors present/summarize the available literature data regarding cytoprotective agents and discuss the role of these agents in preventing damages to cell physiology. The manuscript covers this very important topic in a quite detailed way. It contains 3 figures and 1 table, and is based on 66 publications (24% are publications from the last 5 years). English is satisfactory but needs to be carefully checked. In summary, after minor corrections it can be accepted for publication.

Minor points:

  1. Since the Authors compared synthetic agents with natural ones, the Reviewer suggests the following title “Synthetic and natural cytoprotective agents in cryopreservation: a focus on melatonin”. The adjective “cytoprotective” should be used instead of “cytoprotection”.
  2. In the Abstract the sentence “…the role of cytoprotection agents in preventing cell physiology” is incorrect and should be replaced by “…the role of cytoprotective agents in protecting cell physiology”.
  3. In Figure 1 and in the legend of Figure 1 it should be “phospholipid organization“ instead of “phospholipids organization” and “carbohydrate changes” instead of “carbohydrates changes”.
  4. Abbreviation “CPA” should be introduced in line 43 after “…cytoprotection agents…” but in fact it should be ”..cytoprotective agents..” and not “..cytoprotection agents..”. Also, CPA could be introduced in subtitle So, it should be “2. The cryopreservation process and the role of cytoprotective agents (CPA)”.
  5. In line 78 and 79 the Authors use terms “…penetrating and non-penetrating” and refer to Table 1, in which they use the term “permeating” and “non-permeating”. Although it means roughly the same the Authors should choose and use one term throughout the entire manuscript. The same is with the term DMSO, it makes no sense to use two names, DMSO and Me2SO
  6. In line 149, in case the term “Me2SO” is kept, please correct it.
  7. In line 156 instead of “..CPAs toxicity” should be “..CPA toxicity”.
  8. In line 81 instead of “(eg. Sugars)” should be “(eg. sugars)”.
  9. In line 84 instead of “Non penetrating” should be “Non-penetrating”.
  10. In line 165 “Evolution” should not be in capital letter.
  11. In line 223 in the subtitle “3. Natural Deep eutectic systems” abbreviation “(DESs)“ could be added similarly as it is in subtitle “4.1”; in line 224 instead of “(DES)” should be “(DESs)”.
  12. In line 237 and 239 it should be written “HaCaT” and not “HacaT”.
  13. In Figure 3 above the structural formula it should be written “Melatonin” and in the legend of Figure 3 some abbreviations should be explained. In fact, there are many abbreviations in the manuscript which should be explained, for instance those in lines 271, 279, 338, 343, 359, 371, 386. Thus, an additional paragraph entitled “Abbreviations” should be added to the main body of the manuscript.
  14. In some sentences, the dot is in front of citation but it should be after the cited position, for instance in line 276 or 411.
  15. In line 383 it should be “3 mM” instead of “3mM”.
  16. The terms “in vivo” “in vitro”, “via” should be in italic.
  17. In line 419 instead of “Melatonin inhibits the expression of Bax and increased the expressions of Bcl-2” it should be “Melatonin inhibits the expression of Bax and increases the expression of Bcl-2”.

Reviewer 3 Report

The mini-review article "Natural cytoprotection agents in cryopreservation: a focus on melatonin" evaluated common cytoprotection agents of synthetic and natural origin, describing the alleged targets, mechanisms of protection and side effects. The article is interesting and well written. It can be recommended for publication but it should be structured and prepared carefully and according to the recent findings related to the topic. For example, The artilce main focus is Melatonin but authors have not inroduced it properly in the introduction section. Moreover, the introduction section only cites two articles which is not acceptable. i advise to restrcuture and re-write the introduction by dividing in following sections. 1. cryopreservation 2. Natural cytoprotection agents 3. Meltonin and its importance in this context 4. previous limitations of conduted studies 5. objectives of this review and why it is different from the available studies in context.

The whole MS is divided into small small Paragraphs, some contains 2 lines some 3 or four. this looks bad, please restructure and each paragraph should constiture 115-20 lines instread. Use connecting sentences and your ideas to link your paragraphs rather then providing each paragraph (2-3 lines) and providing citation at the end. 

Conclusion section should be single paragraph ending with the future directions and prospects about the melatonin research in this context.

I only spotted 2 studies cited from 2021 and 3 from 2020, 4 from 2019.  A review article should include recent findings in last 5 years, i suggest remove old data and add recent findings to improve the overall MS.

Reviewer 4 Report

The manuscript ‘’Natural cytoprotection agents in cryopreservation: a focus on melatonin” by Marcantonini et al., required major revision before its acceptance in Molecules.

Authors should kept in mind abstract and include all in Introduction section such as antifreeze proteins, sugar, melatonin etc that is missing from introduction section.

Figure 3 please add the role of arrows shown in upward and downward in figure legend.

Include latest literature in each section that is lacking in the manuscript.

In 4.3. Natural Deep eutectic systems section elaborates the role of osmolytes.

Homogenize the use of terminology throughout the manuscript

Round 2

Reviewer 1 Report

The authors have addressed most of the questions.  However, I still have concerns regarding Table 1.

  1. I still cannot understand what is written inside the brackets. For example, in the case of sulfoxides, dimethyl sulfoxide and methylsulfinylmethane are both the names of the same compound. However, in the second row, the compounds written in the brackets are not different names of the same compound. They are in fact, different compounds. Also, methyl glycol and 1,2-propanediol are not same compounds (written in the Propylene glycol (diols) row). Similar problems can be seen in other rows also.

I feel the table is randomly organized now. In some places, different names of the same compounds are written (In fact, I do not even understand why so many different names need to be provided, unless it is the common name) inside brackets and in other cases, entirely different compounds are mentioned.

  1. Also, I do not believe that the polyampholytes that you cited can be classified as alcohol derivatives.
  2. In the footnote of the Table, what is the rationale behind suddenly mentioning formulations used in drug formulations and food ingredients?
  3. The spelling of synthesized is incorrect in the source column of sulfoxides. It is written as “Synthetized”
  4. I do not believe that the source of methyl glycol is the hydration of propylene oxide. Please double-check check for all the compounds if the source you mentioned is correct.

Author Response

The authors have addressed most of the questions.  However, I still have concerns regarding Table 1.

  1. I still cannot understand what is written inside the brackets. For example, in the case of sulfoxides, dimethyl sulfoxide and methylsulfinylmethane are both the names of the same compound. However, in the second row, the compounds written in the brackets are not different names of the same compound. They are in fact, different compounds. Also, methyl glycol and 1,2-propanediol are not same compounds (written in the Propylene glycol (diols) row). Similar problems can be seen in other rows also.

I feel the table is randomly organized now. In some places, different names of the same compounds are written (In fact, I do not even understand why so many different names need to be provided, unless it is the common name) inside brackets and in other cases, entirely different compounds are mentioned.

Answer: we acknowledge this reviewer for the detailed revision that has helped us a lot to improve this manuscript. We agree that the information provided in brackets cha generate some confusion and an itemized revision of this information is now provided according with the comments of this Reviewer:

  • The information inside the brackets has been removed and essential names and examples of each CPA category have been moved to the second column
  • The term “methylsulfinylmethane” has been eliminated
  • The names of the other rows have been maintained as examples and the fact that they indicate different compounds is what we meant to indicate, i.e. different examples of that category/group of compounds

We hope that this revision may satisfy the requests of this Reviewer, but of course other and more specific suggestion on what we should maintain or eliminate as names in each category/row, will be more than welcome.

  1. Also, I do not believe that the polyampholytes that you cited can be classified as alcohol derivatives.

Answer: the Reviewer is right and this point has been revised according with the info provide in the previous answer

  1. In the footnote of the Table, what is the rationale behind suddenly mentioning formulations used in drug formulations and food ingredients?

Answer: the footnote has been eliminated

  1. The spelling of synthesized is incorrect in the source column of sulfoxides. It is written as “Synthetized”

Answer: this word has been revised accordingly

  1. I do not believe that the source of methyl glycol is the hydration of propylene oxide. Please double-check check for all the compounds if the source you mentioned is correct.

Answer: the sentence and the other sentences referring to the sources if the different CPA have been revised

Reviewer 4 Report

The comments are adequately addressed and is suitable for publication

Author Response

Comments and Suggestions for Authors

The comments are adequately addressed and is suitable for publication

Answer: we acknowledge this reviewer for the revision that has helped us to improve this manuscript.